# Piecewise-Linear Manifolds for Deep Metric Learning

Shubhang Bhatnagar and Narendra Ahuja
University of Illinois Urbana-Champaign
{sb56, n-ahuja @illinois.edu}

Unsupervised deep metric learning (UDML) focuses on learning a semantic representation space using only unlabeled data. This challenging problem requires accurately estimating the similarity between data points, which is used to supervise a deep network. For this purpose, we propose to model the high-dimensional data manifold using a piecewise-linear approximation, with each low-dimensional linear piece approximating the data manifold in a small neighborhood of a point. These neighborhoods are used to estimate similarity between data points. We empirically show that this similarity estimate correlates better with the ground truth than the similarity estimates of current state-of-the-art techniques. We also show that proxies, commonly used in supervised metric learning, can be used to model the piecewise-linear manifold in an unsupervised setting, helping improve performance. Our method outperforms existing unsupervised metric learning approaches on standard zero-shot image retrieval benchmarks.

## 1. Introduction

Deep metric learning (DML) is a challenging yet important task in computer vision, with applications in open-set classification [1, 2], image retrieval [3, 4], few-shot learning [5, 6], and face verification [7, 8]. Deep metric learning aims to learn a representation space with semantically similar data points grouped close together and dissimilar data points located further apart. This involves fine-tuning a pre-trained neural network to minimize a metric learning loss [3, 9–11] on datasets [12–14] with fine-grained class labels available for supervision, which are expensive to obtain. Unsupervised deep metric learning (UDML) intends to learn such a semantic metric space using unlabeled data, enabling us to leverage vast amounts of available unlabeled data without incurring correspondingly large labeling costs.

However, learning a fine-grained semantic space without any labels is very challenging. Current techniques [15–17] focus on using clustering on representations generated from a pre-trained network to estimate similarity between points. But these estimates are often noisy, as the identification of clusters is error-prone, especially when there is a significant domain shift between the dataset used for pre-training (for example, ImageNet [18]) and the one used for metric learning (for example, SOP [14]). For example, a clustering algorithm is likely to group Points A, B and C (located in a high-density region) shown in Figure 1 into a single cluster, marking them as similar.

We propose to mitigate this issue by modeling the data manifold using a piecewise linear approximation, with each piece being formed from a low-dimensional linear approximation of a point's neighborhood. Building such a piecewise linear manifold helps better estimate the similarity between points, as points belonging to a low-dimensional submanifold are likely to be similar due to shared features. For example, in Figure 1, points within linear submanifolds A-E shown are likely to be similar to other points within the same submanifold.

The piecewise linear manifold model enables us to calculate an informative continuous-valued similarity between a pair of points compared to their binary similarity defined by their membership of the cluster. In our model, the similarity between a pair of points $(\mathbf{x}_1, \mathbf{x}_2)$ is inversely proportional to (1) the orthogonal distance $(o_{1,2})$ of point $\mathbf{x}_1$ from the linear neighborhood D of $\mathbf{x}_2$ and vice versa and (2) the distance between point $\mathbf{x}_2$ and the projection of point $\mathbf{x}_1$ on D which are shown in Figure 1. Similarity decays faster orthogonal to a neighborhood than within the neighborhood.

The network is trained to ensure that Euclidean distance in its embedding space between pairs of points is reflective of their dissimilarity estimated using the piecewise linear model using a simple squared error loss function, pushing points together/apart as shown in Figure 1.

First Conference on Parsimony and Learning (CPAL 2024).

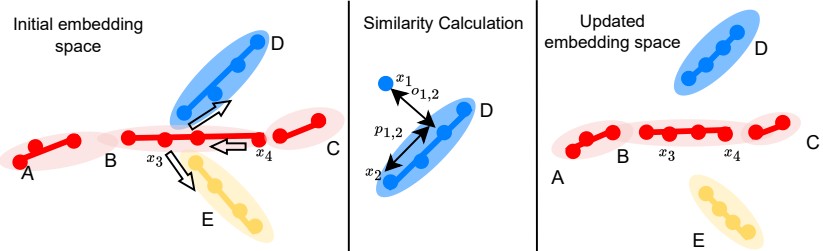

Figure 1: We visualize a 2D data manifold along which data embeddings lie. It may consist of multiple submanifolds, three of which are shown here in red, yellow and blue. Embeddings obtained using pre-trained features (left) are not optimal, with semantically dissimilar points (different colored) being closer than similar points. Our method helps improve the feature space in 3 steps carried out iteratively: **(1)** Identify and approximate the submanifold at each point (eg. Ellipses A-E) by a linear model over aneighborhood small enough to (i) not contain multiple submanifolds and (ii) the contained single submanifold is adequately linear. Such low-dimensional (1D) subspaces are assumed to contain semantically similar points (empirically backed by Sec 4.3). **(2)** Estimate similarity for each pair of points $(\mathbf{x}_1, \mathbf{x}_2)$, in terms of the lengths of projections of vector $\mathbf{x}_1 - \mathbf{x}_2$ on the linear neighborhood D, $p_{1,2}$, and on the normal to D, $o_{1,2}$. Similarity decays faster orthogonal to a neighborhood than along it. **(3)** Train network embedding to bring similar points closer together and push dissimilar points apart. This brings closer together points within the same low-dimensional neighborhoods or those in different neighborhoods but the same low-dimensional (1D) space (eg. A, B, C) closer together. Points not lying in the same low dimensional space (eg. B, D, E) are pushed away from each other.

We propose to make use of proxies to model the piecewise linear manifold beyond what has been sampled in a mini-batch, ensuring better performance. Each proxy is associated with a linear manifold that approximates its neighborhood, and both the proxy location and the orientation of the linear manifold are learned. To the best of our knowledge, we are the first to demonstrate the utility of proxies in such an unsupervised learning framework.

To validate the quality of the semantic space learned by our method, we evaluate it on standard [14, 15, 19] zero-shot image retrieval benchmarks, where it outperforms current state-of-the-art UDML methods by 2.9, 1.5 and 1.3 % in terms of R@1 on the CUB200 [13], Cars-192 [12] and SOP datasets [14], respectively.

To summarize our contributions,

- We present a novel UDML method that constructs a piecewise linear approximation of the data manifold to estimate a continuous-valued similarity between pairs of points.

- We empirically show that our piece-wise linear manifolds enable better identification of points belonging to the same class as compared to a straightforward clustering in the high dimensional space,

- We make use of proxies in modeling the piecewise linear manifold, demonstrating for the first time their utility in UDML.

- We evaluate our method on three standard image-retrieval benchmarks where it outperforms current state-of-the-art techniques.

## 2. Related Work

**Metric Learning:** The goal of deep metric learning is to train a network to learn a semantic feature space from data. Classical approaches include contrastive and triplet losses [9, 10], and their variants [20, 21], which optimize sample distances to bring together examples of the same class while pushing apart those in different classes within a sampled tuplet. Proxy-based approaches [3, 22] attempt to mitigate such tuplet-sampling complexity by instead using a learnable vector to represent points belonging to a class. Sample-sample interactions are substituted by sample-proxy, enabling better, more uniform supervision for each batch of samples and quicker convergence. These methods require fine-grained class labels for learning a semantic feature space, which are expensive to obtain at scale.

**Unsupervised Metric Learning:** Unsupervised metric learning involves using only unlabeled data to learn a semantically meaningful space. Common approaches include pseudo-labeling the data using off-the-shelf clustering algorithms in a pre-trained representation space [17, 23], followed by the application of standard metric learning methods. [15] proposes to use hierarchical clustering to generate pseudo-labels, while [24] uses a random walk for the same. These methods are limited in their ability to accurately model similarity between points due to very high noise in the pseudo-labels generated.

Other approaches [25–27] instead rely on an instance discrimination loss which enforces similarity between different augmentations of an instance. [16] propose a combination of pseudo-labeling and instance discriminative approaches using a Siamese network. [19] introduces a self-training framework with a momentum encoder to continuously improve the quality of pseudo-labels generated using a random walk.

**Self-Supervised Learning:** Self-supervised learning also focuses on learning a representation space in the absence of labels, but the goal is to learn a space that can be fine-tuned effectively (using a small amount of labeled data) for different downstream tasks. Most recent techniques [28–32] for self-supervised learning rely on instance discriminative techniques. The representations learned lack class similarity information, and are less suitable to be directly used for tasks like image retrieval as shown in [19].

**Manifold Learning:** Manifold learning focuses on uncovering low-dimensional structures in high-dimensional data. Manifold learning techniques like LLE [33] and Isomap[34] utilize information derived from the linearized neighborhoods of points to construct low dimensional projections of non-linear manifolds in high dimensional data. Our method also uses linearized neighborhoods of points to construct the piecewise linear manifold. In contrast to other manifold learning techniques, which force a single linear manifold in the neighborhood, our method allows as many manifolds as needed.

# 3. Method

## 3.1. Setup

Given data $\mathcal{D} = \{\mathbf{x}_i\}, i \in \{1 \ldots |\mathcal{D}|\}$, deep metric learning is formulated as training a network $f_\theta$ parameterized by $\theta$ such that the semantic dissimilarity between a pair of samples $\mathbf{x}_1, \mathbf{x}_2 \in \mathcal{D}$ is proportional to the Euclidean distance between their projections $\|f_\theta(\mathbf{x}_1) - f_\theta(\mathbf{x}_2)\|_2$.

After sampling a batch of data $\mathcal{B}$, our method **(1)** Constructs a piecewise linear manifold from them **(2)** Estimates the point-point and proxy-point similarities $s(\mathbf{x}_1, \mathbf{x}_2)$ (calculated for two points $\mathbf{x}_1, \mathbf{x}_2$) using the piecewise linear manifold and **(3)** Trains the network $f_\theta$ and proxies using backpropagation such that Euclidean distances between any two point ($\|\mathbf{x}_1 - \mathbf{x}_2\|_2$) reflects their dissimilarity $1 - s(\mathbf{x}_1, \mathbf{x}_2)$. We go over the details of these steps in the following subsections. Figure 2 provides an overview of our method.

### 3.1.1. Nearest Neighbor Sampling

To enable the construction of a piecewise linear manifold using only the data in a mini-batch, we use nearest neighbor sampling to assemble the mini-batch $\mathcal{B}$. This is due to a randomly sampled batch being less suitable for constructing a piecewise linear manifold because the low-dimensional linear approximation only holds in a small neighborhood of a data point. For large datasets where $|\mathcal{B}| << |\mathcal{D}|$ (common in practice), a randomly sampled batch contains an insufficient number of points lying in low-dimensional linear neighborhoods of each other. So, for constructing a batch, we randomly sample $N$ points followed by their $k - 1$ nearest neighbors, to help better estimate valid linear submanifolds (here $k = \frac{|\mathcal{B}|}{N}$).

### 3.1.2. Momentum Encoder

We maintain an exponential moving average $\phi_t$ of the network's weights -

$$\phi_t = \gamma \phi_t + (1 - \gamma)\theta_t \tag{1}$$

Here $\gamma$ is the momentum parameter, while $\phi_t$, $\theta_t$ are the parameter values at the t-th iteration. The network with the same architecture as $f_\theta$, but weights $\phi$ is referred to as momentum encoder $f_\phi$ here on.

We use representations of the sampled batch $f_\phi(\mathbf{x}_i)i \in \{1 \ldots |\mathcal{B}|\}$ given by the momentum encoder $f_\phi$ as opposed to using the network $f_\theta$, as this helps stabilize the data-manifold. Using a rapidly changing network

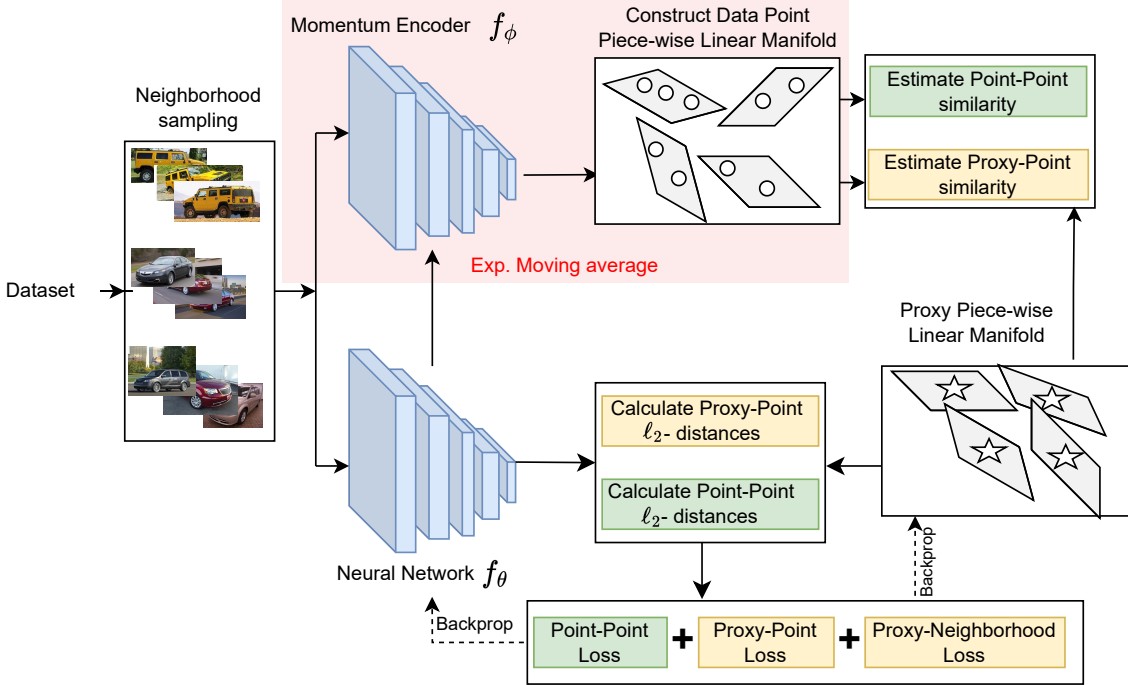

Figure 2: An overview of our method. Points are selected from the dataset using the neighborhood sampling strategy (Sec. 3.1.1), followed by the calculation of their embeddings using the network $f_\theta$ and the momentum encoder $f_\phi$ (Sec. 3.1.2). Embeddings generated by the momentum encoder are used to construct a piecewise linear approximation (Sec. 3.2) of the data manifold. These embeddings are used to calculate point-point (Sec. 3.3) and proxy-point (Sec. 3.5) similarities. The similarities are used to modulate the distance between point-point (Sec.3.6.1) and proxy-point (Sec. 3.6.2) pairs by updating the network $f_\theta$. Locations and neighborhoods of proxies (as described in Sec. 3.4) are also updated using the proxy-point (Sec. 3.6.2) and proxy-neighborhood loss (Sec. 3.6.3) components through backpropagation. Losses colored yellow/green are calculated only using quantities with the same color

$f_\theta$ for constructing and updating the piecewise linear manifold results in unstable training, necessitating the use of a momentum encoder.

## 3.2. Construction of Piecewise-Linear Manifold

To construct a piecewise linear approximation of the data manifold, we calculate an $m$-dimensional linear submanifold $P_i$ at each point $f_\phi(\mathbf{x}_i) \in \mathcal{B}$ to approximate the data manifold in its neighborhood. The linear submanifold $P_i$ should be small enough such that the linear approximation holds, but large enough to fit any points which lie on it. We use PCA to calculate a low-dimensional linear approximation of the point's neighborhood. The PCA is applied to a subset of its $k$ nearest neighbors. Henceforth, we refer to the j-th nearest neighbor of $f_\phi(\mathbf{x}_i)$ as $\mathcal{N}(f_\phi(\mathbf{x}_i))_j, j \in \{1 \ldots k\}$. A subset of the k-nearest neighbors is used because not all points in its neighborhood might fit well on the linear submanifold (also, the model is constantly updating). An iterative algorithm described below is used to pick the subset of points $\mathcal{X}_i$ which are used to find the best fit linear submanifold

1. Pick the point $f_\phi(\mathbf{x}_i)$ and its $m - 1$ nearest neighbors, $\mathcal{N}(f_\phi(\mathbf{x}_i))_j, j \in \{1 \ldots \mathcal{P} - 1\}$ to form a set of points $\mathcal{X}_i$. An $m$ dimensional linear submanifold constructed by applying PCA to $\mathcal{X}_i$ would fit all points with zero error.

2. Construct a set $\mathcal{X}_i' = \mathcal{X}_i \cup \mathcal{N}(f_\phi(\mathbf{x}_i))_m$ by adding the $m$-th nearest neighbor of $f_\phi(\mathbf{x}_i)$ to $\mathcal{X}_i$. Construct the best fit $m$ dimensional submanifold by applying PCA on $\mathcal{X}_i'$.

3. If the $m$ dimensional submanifold can reconstruct all points $\in \mathcal{X}_i$ with an error less than threshold $T\%$, we update $\mathcal{X}_i = \mathcal{X}_i'$. If the error for any point is greater than $T\%$, it implies that the newly added point $\mathcal{N}(f_\phi(\mathbf{x}_i))_m$ is not a good fit to the linear manifold, and is not added to $\mathcal{X}_i$.

4. Steps 2 and 3 are repeated for all other points $N(f_\phi(\mathbf{x}_i))_j, j \in \{m+1 \ldots k\}$ in the neighborhood of $f_\phi(\mathbf{x}_i)$, in ascending order of their distance from $f_\phi(\mathbf{x}_i)$.

The set $\mathcal{X}_i$ so obtained contains all points in the neighborhood of $f_\phi(\mathbf{x}_i)$ that fit in an $m$-dimensional linear submanifold. A basis for the subspace containing $P_i$ is formed by using PCA on $\mathcal{X}_i$ followed by selecting the top m eigenvectors (best fit). In the subset selection algorithm, the threshold $T$ and the dimension $m$ are hyperparameters fixed empirically.

## 3.3. Estimating Point-Point Similarity

Using the piecewise linear approximation of the data manifold at each point $f_\phi(\mathbf{x}_i) \in \mathcal{B}$, we calculate the similarity $s^{'}(\mathbf{x}_i, \mathbf{x}_j)$ between any two points $\mathbf{x}_i, \mathbf{x}_j$. We model the as a product of two components

$$s^{'}(\mathbf{x}_i, \mathbf{x}_j) = \alpha(\mathbf{x}_i, \mathbf{x}_j)\beta(\mathbf{x}_i, \mathbf{x}_j) \tag{2}$$

$\alpha(\mathbf{x}_i, \mathbf{x}_j)$ is based on distance between $f_\phi(\mathbf{x}_i)$ and $f_\phi(\mathbf{x}_j)$ projected on the normal to $P_j$ (the linear submanifold neighborhood of $f_\phi(\mathbf{x}_j)$) and, $\beta(\mathbf{x}_i, \mathbf{x}_j)$ which takes into consideration their distance projected on $P_j$. Factorizing the similarity into these components helps quantify the differing roles of the orthogonal distance and projected distance (Section 4.4.3). Figure 1 illustrates these distances for a 2D toy example.

Functional forms for $\alpha(\mathbf{x}_i, \mathbf{x}_j)$ and $\beta(\mathbf{x}_i, \mathbf{x}_j)$ are chosen such that similarity decays with an increase in both these distance components. We choose $\alpha(\mathbf{x}_i, \mathbf{x}_j)$ as:

$$\alpha(\mathbf{x}_i, \mathbf{x}_j) = \frac{1}{(1 + \frac{o(\mathbf{x}_i, \mathbf{x}_j)}{2})^{N_\alpha}} \tag{3}$$

where $o(\mathbf{x}_i, \mathbf{x}_j)$ is the distance between $f_\phi(\mathbf{x}_i)$ and $f_\phi(\mathbf{x}_j)$ projected on the normal to $P_j$. The choice of constants ensures that $\alpha(\mathbf{x}_i, \mathbf{x}_j) \in [0, 1]$ for $o_{i,j} \in [0, 2]$ (which is true as all embeddings $f_\phi(x)$ have unit norm). $N_\alpha$ controls the sharpness of similarity decay. Similarly, we choose $\beta(\mathbf{x}_i, \mathbf{x}_j)$ as:

$$\beta(\mathbf{x}_i, \mathbf{x}_j) = \frac{1}{(1 + p(\mathbf{x}_i, \mathbf{x}_j))^{N_\beta}} \tag{4}$$

where $p(\mathbf{x}_i, \mathbf{x}_j)$ is the distance between $f_\phi(\mathbf{x}_i)$ and $f_\phi(\mathbf{x}_j)$ projected on $P_j$. The constants ensure $\beta(\mathbf{x}_i, \mathbf{x}_j) \in [0, 1]$ for $p(\mathbf{x}_i, \mathbf{x}_j) \in [0, 1]$. $N_\beta$ controls the sharpness of the decay in similarity with $p(\mathbf{x}_i, \mathbf{x}_j)$. After calculating the similarity $s^{'}(\mathbf{x}_i, \mathbf{x}_j)$, we similarly calculate $s^{'}(\mathbf{x}_j, \mathbf{x}_i)$ using $P_j$, the linear submanifold in the neighborhood of $f_\phi(\mathbf{x}_j)$. We calculate the average similarity $s(\mathbf{x}_i, \mathbf{x}_j)$ as

$$s(\mathbf{x}_i, \mathbf{x}_j) = \frac{s^{'}(\mathbf{x}_i, \mathbf{x}_j) + s^{'}(\mathbf{x}_j, \mathbf{x}_i)}{2} \tag{5}$$

The average similarity $s(\mathbf{x}_i, \mathbf{x}_j)$ is symmetric that is $s(\mathbf{x}_i, \mathbf{x}_j) = s(\mathbf{x}_j, \mathbf{x}_i)$, unlike $s^{'}(\mathbf{x}_i, \mathbf{x}_j)$. In our experiments, we use $N_\alpha > N_\beta$ to impose a more severe penalty on points that do not fit in the low-dimensional linear submanifold $P_i$ than points on the submanifold $P_i$ which are located away from $f_\phi(\mathbf{x}_i)$.

## 3.4. Proxies to Model Manifold

When training on most real datasets, $|\mathcal{B}| << |\mathcal{D}|$, which means that the linear approximations of point neighborhoods $P_i \forall i \in \{1, \ldots, |\mathcal{B}|\}$ might not be able to effectively represent the whole data manifold. To mitigate this issue, we use proxies to model the data manifold. Proxies have been used to model the data distribution in supervised deep metric learning [3, 35], where they are typically used as learnable cluster centers belonging to a class. In the unsupervised setting, a similar, direct use is not possible in the absence of any label information.

We propose to instead use proxies to model low-dimensional linear approximations of point neighborhoods on the data manifold. Specifically, each proxy $\boldsymbol{\rho}_j \forall j \in \{1 \ldots N_{\boldsymbol{\rho}}\}$ is associated with $\Psi_j$, a set of m-orthonormal vectors $(\boldsymbol{\psi}_{1,j} \ldots \boldsymbol{\psi}_{m,j})$ representing the $m$-dimensional linear approximation of the proxy's neighborhood. Here, $N_{\boldsymbol{\rho}}$ represents the number of proxies chosen to be used. Each proxy $\boldsymbol{\rho}_j$ along with its linear neighborhood $\Psi_j$ represents a piecewise-linear approximation to a part of the data manifold. This helps model the similarity of a point $f_\phi(\mathbf{x}_i)$ with a larger part of the data manifold than that represented by only the batch. The proxies $\boldsymbol{\rho}_j$ and their linear neighborhoods $\Psi_j$ are learnable parameters, updated in the backward pass along with model parameters.

## 3.5. Estimating Proxy-Point Similarity

The similarity $s(f_\phi(\mathbf{x}_i), \boldsymbol{\rho}_j)$ between a point $f_\phi(\mathbf{x}_i)$ and a proxy $\boldsymbol{\rho}_j$ is estimated using Equation 5 in the same manner as described in Section 3.3. In the similarity calculation described in Section 3.3, $f_\phi(\mathbf{x}_j)$ is replaced with $\boldsymbol{\rho}_j$, with the vectors $\Psi_j$ playing the role of the point neighborhood $P_j$ required in the process.

## 3.6. Loss & Training

We design our loss function to ensure that 1) The Euclidean distance between any pair of points or a proxy-point pair is proportional to the measured dissimilarity between them and 2) The proxies and their neighborhoods $\Psi_j$ are updated continuously. This is ensured using 3 components 1)**Point-Point loss** 2)**Proxy-Point loss** 3)**Proxy-Neighborhood loss**

### 3.6.1. Point-Point Loss

The point-point interaction component of the loss $\mathcal{L}_{point}$ modulates the Euclidean distance between a given pair of points using the estimate of their similarity. Specifically, for points $\mathbf{x}_i, \mathbf{x}_j$, the loss is the square of the difference between their Euclidean distance $\|f_\theta(\mathbf{x}_i) - f_\theta(\mathbf{x}_j)\|_2$ and the estimated dissimilarity $1 - s(\mathbf{x}_i, \mathbf{x}_j)$ multiplied by a scaling factor $\delta$. This is summed over all pairs in the batch to calculate $\mathcal{L}_{point}$ given by

$$\mathcal{L}_{point} = \sum_{i=1}^{|\mathcal{B}|} \sum_{j=1, j \neq i}^{|\mathcal{B}|} \left( \delta \times (1 - s(\mathbf{x}_i, \mathbf{x}_j)) - \|f_\theta(\mathbf{x}_i) - f_\theta(\mathbf{x}_j)\|_2 \right)^2 \tag{6}$$

The loss ensures that points that are very similar with $s(\mathbf{x}_i, \mathbf{x}_j)) \to 1$ are attracted to each other to have $\ell_2$ distance $\to 0$. Points dissimilar to each other with $s(\mathbf{x}_i, \mathbf{x}_j)) \to 0$ are repelled from each other until their $\ell_2$ distance $\to \delta$.

### 3.6.2. Proxy-Point Loss

The proxy-point interaction component of the loss $\mathcal{L}_{proxy}$ is also based on squared error and is similar to the point-point component with the only difference being the replacement of point-point similarity with proxy-point similarity. It is given by

$$\mathcal{L}_{proxy} = \sum_{i=1}^{|\mathcal{B}|} \sum_{j=1}^{N_{\boldsymbol{\rho}}} \left( \delta \times (1 - s(\mathbf{x}_i, \boldsymbol{\rho}_j)) - \|f_\theta(\mathbf{x}_i) - \boldsymbol{\rho}_j\|_2 \right)^2 \tag{7}$$

The $\delta$ is the same as the one used in the point-point component.

### 3.6.3. Proxy-Neighborhood Loss

The proxy-neighborhood $\mathcal{L}_{neighborhood}$ component ensures that the proxy neighborhoods $\Psi_j$ are updated as the proxies move. It is also formulated as a squared error, helping align proxy neighborhoods ($\boldsymbol{\psi}_{k,j}$, the basis vectors used to define them) with those of points most similar to them. Specifically, $\mathcal{L}_{neighborhood}$ is defined as

$$\mathcal{L}_{neighborhood} = \sum_{i=1}^{|\mathcal{B}|} \sum_{j=1}^{N_{\boldsymbol{\rho}}} \sum_{k=1}^{m} \left( s(\mathbf{x}_i, \boldsymbol{\rho}_j) - cos\theta_{k,j,i} \right)^2$$
$$\text{where } \theta_{k,j,i} = \angle \text{ between } \boldsymbol{\psi}_{k,j} \text{ and } P_i \tag{8}$$

$cos\theta_{k,j,i}$ quantifies the similarity between $\boldsymbol{\psi}_{k,j}$ and linear submanifold $P_i$.

### 3.6.4. Training

The combined loss $\mathcal{L}$ is defined as the sum of its three components -

$$\mathcal{L} = \mathcal{L}_{point} + \mathcal{L}_{proxy} + \mathcal{L}_{neighborhood} \tag{9}$$

We calculate the loss $\mathcal{L}$ for a batch $\mathcal{B}$ and update the network parameters $\theta$ via mini-batch gradient descent. $\phi$ is instead updated using Eq. 1. Note that network parameters $\theta$ are only affected by $\mathcal{L}_{point}$, $\mathcal{L}_{proxy}$, while the proxies $\boldsymbol{\rho}$ and their neighborhoods $\Psi$ are only affected by $\mathcal{L}_{proxy}$, $\mathcal{L}_{neighborhood}$.

| Benchmarks → | CUB-200-2011 | | | | Cars-196 | | | | SOP | | |
|---|---|---|---|---|---|---|---|---|---|---|---|
| Methods ↓ | R@1 | R@2 | R@4 | R@8 | R@1 | R@2 | R@4 | R@8 | R@1 | R@10 | R@100 |
| **GoogleNet (128 dim)** | | | | | | | | | | | |
| Examplar [38] | 38.2 | 50.3 | 62.8 | 75.0 | 36.5 | 48.1 | 59.2 | 71.0 | 45.0 | 60.3 | 75.2 |
| NCE [25] | 39.2 | 51.4 | 63.7 | 75.8 | 37.5 | 48.7 | 59.8 | 71.5 | 46.6 | 62.3 | 76.8 |
| DeepCluster [17] | 42.9 | 54.1 | 65.6 | 76.2 | 32.6 | 43.8 | 57.0 | 69.5 | 34.6 | 52.6 | 66.8 |
| MOM [24] | 45.3 | 57.8 | 68.6 | 78.4 | 35.5 | 48.2 | 60.6 | 72.4 | 43.3 | 57.2 | 73.2 |
| AND [39] | 47.3 | 59.4 | 71.0 | 80.0 | 38.4 | 49.6 | 60.2 | 72.9 | 47.4 | 62.6 | 77.1 |
| ISIF [26] | 46.2 | 59.0 | 70.1 | 80.2 | 41.3 | 52.3 | 63.6 | 74.9 | 48.9 | 64.0 | 78.0 |
| sSUML [40] | 43.5 | 56.2 | 68.3 | 79.1 | 42.0 | 54.3 | 66.0 | 77.2 | 47.8 | 63.6 | 78.3 |
| Ortho [41] | 47.1 | 59.7 | 72.1 | 82.8 | 45.0 | 56.2 | 66.7 | 76.6 | 45.5 | 61.6 | 77.1 |
| PSLR [42] | 48.1 | 60.1 | 71.8 | 81.6 | 43.7 | 54.8 | 66.1 | 76.2 | 51.1 | 66.5 | 79.8 |
| ROUL [36] | 56.7 | 68.4 | 78.3 | 86.3 | 45.0 | 56.9 | 68.4 | 78.6 | 53.4 | 68.8 | 81.7 |
| SAN [43] | 55.9 | 68.0 | 78.6 | 86.8 | 44.2 | 55.5 | 66.8 | 76.9 | 58.7 | 73.1 | 84.6 |
| STML* [19] | 57.7 | 69.8 | 80.1 | 87.1 | 48.0 | 58.7 | 69.5 | 79.5 | 63.8 | 77.8 | 88.9 |
| **Ours** | **60.6 ± 0.3** | **71.1 ± 0.2** | **81.1 ± 0.1** | **87.8 ± 0.1** | **49.5 ± 0.3** | **60.6 ± 0.3** | **72.1 ± 0.2** | **80.9 ± 0.2** | **65.1 ± 0.3** | **80.4 ± 0.2** | **90.2 ± 0.1** |
| **GoogleNet (512 dim)** | | | | | | | | | | | |
| UDML-SS [23] | 54.7 | 66.9 | 77.4 | 86.1 | 45.1 | 56.1 | 66.5 | 75.7 | 63.5 | 78.0 | 88.6 |
| TAC-CCL [16] | 57.5 | 68.8 | 78.8 | 87.2 | 46.1 | 56.9 | 67.5 | 76.7 | 63.9 | 77.6 | 87.8 |
| UHML [15] | 58.9 | 70.6 | 80.4 | 87.7 | 47.7 | 58.9 | 70.3 | 80.3 | 65.1 | 78.2 | 88.3 |
| STML* [19] | 58.6 | 70.2 | 80.9 | 87.9 | 48.6 | 60.4 | 71.3 | 80.8 | 65.1 | 79.7 | 89.1 |
| **Ours** | **61.7 ± 0.3** | **72.5 ± 0.2** | **82.2 ± 0.2** | **88.3 ± 0.1** | **51.2 ± 0.2** | **62.2 ± 0.2** | **72.1 ± 0.2** | **81.0 ± 0.1** | **66.4 ± 0.2** | **81.1 ± 0.1** | **90.6 ± 0.1** |

Table 1: Comparison of the Recall@$K$ (%) achieved by our method on the CUB-200-2011, Cars-196 and SOP datasets with state-of-the-art baselines under standard settings described in Section 4.1. The table reports performance and standard deviations of our method calculated over 5 runs. * denotes results reproduced by us under the same settings.

# 4. Experiments and Results

## 4.1. Setup

**Datasets:** We empirically validate our method and compare it with current state-of-the-art baselines on three public benchmark datasets for image retrieval - **(1)** The Cars-196 dataset [12] with 16,185 images belonging to 196 different categories based on the model of the cars **(2)** CUB-200-2011 dataset [13] having 11,788 images 200 classes of birds and **(3)** Stanford Online Products (SOP) dataset [14] having 120,053 images of 22,634 different kinds of products sold online. We use examples belonging to the first half of classes for training while using the other half classes for testing following the commonly used setting previously used by [14, 19, 36] to test zero-shot image retrieval.

**Backbone:** The choice of backbone plays an important role in the image retrieval performance, with stronger backbones yielding significantly better results. For a fair comparison with previous work, we performed experiments using the GoogLeNet backbone [37] commonly used by them. We used embedding sizes of 128 and 512 to enable comparison with all previous work reporting results for either setting. We initialize these networks with ImageNet pre-trained weights unless specified otherwise. We $\ell_2$ normalize the final output of these networks, as is commonly done.

**Training parameters:** We train our network for 200 epochs on all three datasets. We use Adam with the learning rate chosen as $5e^{-4}$. We scale up the learning rate of our proxies by 100 times for faster convergence as is done commonly [22]. For training, we use $227 \times 227$ sized center crops from images after they are resized to $256 \times 256$ as is done commonly [15, 19]. We choose $N_\alpha = 4$ and $N_\beta = 0.5$ in all our experiments, and $N_\rho = 100$. $\delta$ is set as 2 (the maximum distance between two points on a unit sphere) while fixing $m = 3$. The reconstruction quality threshold $T$ is set as $90\%$, while the momentum parameter $\gamma$ is fixed as 0.999. We use a mini-batch size of 100 samples, with each batch having two random augmentations of an image as proposed in [19].

**Evaluation Settings:**. We measure image retrieval performance using the Recall@K metric, which computes the percentage of samples that have a valid similar neighbor (belonging to the same class) among its K nearest neighbors. All experiments are performed on a single NVIDIA V100 GPU.

## 4.2. Image Retrieval Performance

We compare the performance of our method with recent state-of-the-art unsupervised deep metric learning techniques on the three standard benchmarks described before. As seen in Table 3, our method achieves state-of-the-art performance irrespective of the backbone and embedding dimension used. In particular, it outperforms STML[19], the current state-of-the-art by 2.9%, 1.5% and 1.3% in terms of R@1 on the CUB-

| Metric → | Label Purity | | | Correlation with Ground truth | | |
|---|---|---|---|---|---|---|
| Methods ↓ | CUB200 | Cars196 | SOP | CUB200 | Cars196 | SOP |
| K-Means [17] | 0.38 | 0.29 | 0.32 | 0.37 | 0.21 | 0.32 |
| Hierarchical Clustering [15] | 0.49 | 0.33 | 0.39 | 0.52 | 0.36 | 0.49 |
| Random Walk [24] | - | - | - | 0.45 | 0.26 | 0.42 |
| **Ours** | **0.67** | **0.45** | **0.62** | **0.61** | **0.45** | **0.67** |

Table 2: A comparison of the quality of supervision provided by our method as compared to recent pseudo-labeling techniques. Our method is not only better at grouping points of the same class together as evidenced by higher label purity, but also helps estimate a better similarity more correlated to ground truth similarity between two points.

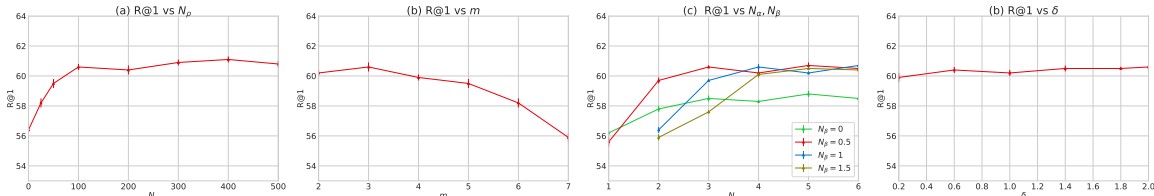

Figure 3: Variation in Recall@1 with $N_\rho$, $m$, $N_\alpha$, $N_\beta$ and $\delta$ on the CUB-200-2011 dataset when using a 128 dim embedding GoogLeNet backbone. Error bars represent standard deviations over 5 runs.

200, Cars-196 and SOP datasets respectively when using a GoogLeNet backbone (128-dim embedding). It similarly outperforms UHML[15], another state-of-the-art baseline (512-dim case) by $2.8\%$, $3.5\%$ and $1.3\%$ in terms of R@1 on the CUB-200, Cars-196 and SOP datasets respectively when using a GoogleNet backbone (512-dim embedding). Note that these improvements are significant compared to previous improvements on these benchmarks. These results point to the wide applicability of using a piecewise linear manifold to model data manifolds and quantify pairwise similarity.

### 4.3. Quality of Supervision using Low-Dimensional Linear Neighborhoods

Our use of low dimensional linear manifolds to model point neighborhoods was motivated by two reasons (1) We hypothesize that they have better purity (in terms of class labels) as compared to clusters obtained using off-the-shelf techniques and (2) They enable calculation of a similarity between any two points which better reflects ground truth.

We empirically test these by comparing the label purity of linear manifolds $P_i$ constructed using our algorithm (Section 3.2) with the label purity of clusters obtained using commonly used clustering algorithms. We perform the comparison on embeddings of the CUB-200 test set obtained using a 128-dim ImageNet pre-trained GoogLeNet model (described in Section 4.1). As seen in Table 2, linear manifolds have significantly better purity, helping better discover underlying class structures.

To calculate the quality of supervision provided by our similarity estimate, we calculate its correlation with the ground truth similarity between any two points. The ground truth similarity is binary, being 1 for two points belonging to the same class and 0 otherwise. Such a binary ground truth similarity is commonly used in DML when labels are available. Unsupervised DML strategies that rely on generating pseudo-labels [15, 17, 24] also calculate estimates of the ground truth similarity for each pair of points using pseudo labels obtained through clustering / random walk. To enable a comparison with our method, we calculate a similar correlation for their similarity estimates obtained using pseudo-labels. As seen in Table 2, our continuous-valued similarity calculated using a linear approximation to neighborhoods of points has a higher correlation with ground truth. This validates the design of our method, showing its wide utility.

### 4.4. Ablation Study & Analysis

In this section, we analyze the role of different components of our method via ablations. We train GoogLeNet models with a 128-dimensional embedding on the CUB-200-2011 dataset using the parameters in Section 4.1, unless specified otherwise. We train a new model from scratch for each separate set of parameters. We measure the performance achieved by the model using Recall@1 (R@1).

### 4.4.1. Effect of $N_\rho$

The number of proxies $N_\rho$ is an important parameter, as the proxies help model relations of sampled data with data outside the batch. We vary it between $[0, 500]$ to understand its effect on performance. The results, plotted in Figure 3(a) show that the performance remains stable for a wide range of $N_\rho$, decreasing when the number of proxies is very small. The cause of this decline is the number of proxies not being large enough to model the underlying data manifold effectively.

### 4.4.2. Effect of linear manifold dimension $m$

The parameter $m$ is the dimension of the linear submanifold $P_i$ which approximates the data manifold in the neighborhood of a point $f_\phi(\mathbf{x}_i)$. We vary $m$ between $[2, 7]$ to understand its role in our method. As seen in Figure 3(b), performance remains stable for small $m$ followed by a decrease as $m$ becomes larger. This is because $P_i$ is used to approximate the immediate neighborhood of a point which is likely to be low dimensional. Using a large $m$ might lead to overfitting, as there are only a limited number of close neighbors of a point available in a batch to estimate $P_i$ leading to performance deterioration.

### 4.4.3. Role of $N_\alpha, N_\beta$

$N_\alpha$ and $N_\beta$ control the decay in similarity based on the orthogonal and projected distance of a point from the linear submanifold in the neighborhood of the other point. We vary $N_\beta$ between $\{0, 0.5, 1, 1.5\}$ and $N_\alpha$ between $[1, 6]$. We train a separate network for each possible $N_\alpha > N_\beta$ pair in this range.

The results of these experiments are visualized in Figure 3(c). We observe three trends (1) Performance (R@1) remains stable for a wide range of $N_\alpha, N_\beta$ when $N_\alpha >> N_\beta$ (2) As $N_\alpha \to N_\beta$, we observe a significant deterioration in performance. This is because when $N_\alpha = N_\beta$, a point A lying at a distance $\epsilon$ in the linear neighborhood of another point B (and hence likely sharing many common features with B and its neighbors) would be judged to be as dissimilar to B as a third point C located at an orthogonal distance of $\epsilon$ from the linear neighborhood of B. Note that we do not plot performance for cases where $N_\alpha \leq N_\beta$ for clarity, as we observed a significant degradation in performance for these cases ($\approx 10\%$ for $N_\alpha = N_\beta$ case). (3) We observe a degradation in performance when $N_\beta = 0$, reflecting its importance. To summarize, performance is the best when data similarity decays faster with orthogonal distance (features alien to the neighborhood) and slower along projected distance (features known to capture similarity).

### 4.4.4. Effect of $\delta$

We conduct an empirical study to determine the effect of varying the distance scale parameter $\delta$ between $[0.2, 2]$. Figure 3(d) plots variation in performance with $\delta$ where we observe that performance remains relatively stable for a wide range of values, demonstrating the robustness of our method.

Further ablation studies demonstrate the importance of (1) our algorithm to construct a piece-wise linear manifold (Sec 3.2) and (2) our proposed similarity functions (Sec. 3.3 & 3.5), details of which can be found in Appendix A.

## 5. Conclusion

We present a novel method to learn a semantically meaningful representation space in the absence of labeled data. Our method constructs a piecewise linear approximation of the data manifold by modeling point neighborhoods as linear manifolds. It uses these linearized neighborhoods to quantify similarity between pairs of points which we show empirically to be more informative than similarity estimated by common clustering-based approaches. We augment the manifold estimated using sample points in a batch with learnable proxies, demonstrating their utility for unsupervised metric learning.

We evaluate the semantic representation of our method on three standard image retrieval tasks where it outperforms current state-of-the-art methods. We hope insights gained from our works motivate further investigation into the structure of data manifolds learned by neural networks and their exploitation for unsupervised representation learning.

## 6. Acknowledgement

The support of the Office of Naval Research under grant N00014-20-1-2444 and of USDA National Institute of Food and Agriculture under grant 2020-67021-32799/1024178 are gratefully acknowledged.

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

# A. Appendix: Ablation on Similarity Calculation

We perform three ablation studies demonstrating the importance of components our method used in similarity calculation: 1) Forming Piece-wise linear manifold (Sec 3.2) 2) Proposed Similarity functions: Sec. 3.3 & 3.5

| Benchmarks → | CUB-200-2011 | |
|---|---|---|
| Methods ↓ | R@1 | R@2 |
| **GoogleNet (512 dim)** | | |
| (1) Ours - PL manifold (Sec 3.2) | 58.4 | 70.8 |
| (2) Ours - similarity (Sec 3.3, 3.5) | 54.2 | 66.5 |
| (3) Ours - PL manifold (Sec 3.2) - similarity (Sec 3.3, 3.5) | 53.1 | 65.4 |
| (4) **Ours** | **61.7** | **72.5** |

Table 3: Comparison of the Recall@$K$ (%) achieved by ablations of our method on the CUB-200-2011 under standard settings described in Section 4.1. The table reports performance and standard deviations of our method calculated over 5 runs.

For all three studies, we use parameters specified in Section 4.1 unless specified otherwise.

In ablation (1), instead of using our algorithm described in Section 3.2 to form neighborhoods by selecting those within a linear submanifold, we form point neighborhoods using all k nearest neighbors of a point. The 3.3% drop in R@1 for this ablation demonstrates the importance of the appropriate selection of piecewise linear neighborhoods using our algorithm.

In ablation (2), instead of calculating similarity using Equation 5, we assign binary similarity values (1 for points in the neighborhood, 0 for those outside) to demonstrate the utility of similarity $s_{i,j}$ (Section 3.3, 3.5). A drop of 7.2 % R@1 provides additional evidence of the design of our distance similarity.

In ablation (3), we remove both the above-mentioned components, using only binary similarity using neighborhoods formed by k nearest neighbors. The 8.6 % drop in R@1 points to the importance of both components of our method.

