# OpenReview forum: "Piecewise-Linear Manifolds for Deep Metric Learning"
_CPAL.cc/2024/Conference — CPAL 2024 (Proceedings Track) Oral_

### Official Review · Reviewer_3LmT · 2023-10-03
**Accept with several questions/suggestions**

**Rating:** 8
**Confidence:** 3

**Review:**

The authors present a novel method for unsupervised deep metric learning by representing data with piecewise linear manifolds. This simple representation allows neighborhoods to be linearly approximated.

Pros:
+ Provides good intuition for their approach
+ Clear motivation
+ Validated their approach against numerous other methods

Cons:
- Method description is fairly long and involves many variables, so an algorithm/pseudocode block would enhance the clarity.
- Limits of their work not explicitly stated
- Choice of GoogLeNet, a fairly old model, over other models is not explained

Questions:

1) The loss function is a sum of three components (point, proxy, and neighborhood). Is there a reason to weight these the same i.e. would performance improve by adding the flexibility to weight them differently?

2) What is the reasoning behind choosing GoogLeNet for your experiments? How does the method perform with different architectures and larger models?

---

### Official Review · Reviewer_vgyP · 2023-10-04
**Good results, but convoluted method**

**Rating:** 5
**Confidence:** 4

**Review:**

This paper proposed a new algorithm for deep metric learning based on estimating the piecewise-linear local manifold. Nearest-neighbor based sampling and proxy vectors are used to mitigate the issue that a random batch doesn’t contain enough samples to form a local manifold. Compared to other metric learning methods, the proposed method achieves significantly better performance.

Overall, this paper is clear and well written. The method and experimental results are clearly presented. The proposed method is motivated in terms of encouraging sample points to come closer to the local manifold formed by neighboring points, but the resulting loss function seems much more complicated than it needs to be. First, there’s very little motivation provided for the particular form of similarity metric s proposed in Section 3.3, besides transforming the projected distance and orthogonal distance to the required numerical range. There’s also no motivation as of why ɑ and ꞵ needs to be multiplied together in equation (2).

It is also interesting that in the actual experiment, the authors mixed this distance term with other simpler terms like L2 distance and cosine similarity again. This happens in equation (6) (7) and (8). A δ parameter is included to adjust the importance of similarity s and other terms. However, in Figure 3, when δ decreases, it only has a very minor effect on the accuracy result, indicating that the other, simpler terms are doing most of the heavy lifting.

The proposed method also contains many extra implementation details that other metric learning methods do not have, for example, using nearest neighbor sampling to form a batch. In the self-supervised learning literature, it is well known that this alone can provide significant performance improvement[1]. Therefore, it will be necessary to study what design choice contributed to the improved performance and by how much.

Although it is nice that the authors achieved significant improvements in their benchmark results, it is at the expense of many more hyper-parameters involved, as well as a more convoluted and ill-motivated design. It would be more interesting to see experiments and analysis about how this improvement is achieved and what principle it reveals.

[1] Debidatta, Dwibedi. Et al. With a Little Help from My Friends: Nearest-Neighbor Contrastive Learning of Visual Representations. arXiv:2104.14548v1

---

### Official Review · Reviewer_cHmX · 2023-10-07
**Good result, more qualitative and exploratory analysis would be helpful.**

**Rating:** 5
**Confidence:** 2

**Review:**

Summary: The paper is dealing with unsupervised deep metric learning, which aims to learn a metric, such that under this representation, point that are semantically closed together are closed together under the learnt metric space. In the work, the author fits a piece-wise linear embedding model on the data, and then propose the metric should follows several criterion based on this piece-wise linear embedding model. Lastly, the author train a neural network to transform the data, under the transformation, the L2 distance should satisfied the proposed criterion. The author assumes there exists a metric in the raw signal space that behaves locally consistent with the metric we want to learn. This allows us to find a neighborhood such that points in this neighborhood is semantically closed to each other. [I believe the author first use a pretrained neural network to extract feature from each image, then perform unsupervised deep metric learning on these features.] The author also assumes the neighborhood is locally low rank which can be approximated with a linear embedding.

Comments:
My background is in signal processing and unsupervised learning. I am not entirely update to date with the work on deep metric learning. My comments will be based on my summary. If anything is incorrect, please let me know, especially the part the bracket.

I think the proposed method seems to work well as it performs bette than existing benchmark. The resulting clustering from each piecewise linear manifold seems to perform better than other clustering algorithm. But I personally would like to see more qualitative and exploratory analysis than just the benchmark result. For example:
1. If the method is called piecewise-linear manifolds, can I visualize the piecewise-linear manifold structure? If the manifold of natural image is still very high dimensional to visualize, can the author to make a toy example for understanding and sanity check that the model indeed works as epected?
2. I wish I have a better understanding on what $o_{i,j}$ and $p_{i,j}$ is because they seems to be crucial. For my understanding, one is the distance between $x_j$ to $x_i$, the other one is the distance between $x_j$ to the manifold $x_i$ belongs to. Seem like these two distances needs to be balanced for the model to have good performance. But I don't get the intuition. The explanation in 4.4.3 still seem to obscure to me.
3. Is the local neighborhood indeed low-rank? The author seems to construct the neighbor by expanding if until it cannot be approximated by a linear subspace. I would like to see a more analysis on this. Like how much can you expand the neighborhood and how the eigenvalue of PCA fall off.
4. I think it would be nice if the author can compare their methods with other manifold learning method. The general assumption and goal is that the metric in the signal space works well locally but not globally. This seems to fit "think globally, fit locally" manifold learning very well.
5. In fact, I think the model can be consider a manifold learning model.  The definition of the "Point-Point," "Proxy-Point" and "Proxy-Neighborhood" similarity together defined a similarity kernel, which is being approximated by the trained neural network. Does the author consider of not using the neural network? Since the signal space is already the feature of a pretrained neural network, maybe the similarity kernel can just be just approximated by a linear transformation. In this case, the model is a specific case of laplacian eigenmap. Or maybe the author should do some ablation study to see if a light weighted neural network will do a similar job, as this will fit the premise of the conference "Parsimony" better.

Saul, L and Roweis, S. Think globally, fit locally: unsupervised learning of low dimensional manifolds.
Tenenbaum, J. et al. A global geometric framework for nonlinear dimensionality reduction.

---

### Meta-Review · Area_Chair_Jxjx · 2023-11-13

**Recommendation:** Accept (Poster)
**Confidence:** 4

**Metareview:**

In this paper, the researchers delve into the problem of learning semantically meaningful representations in an unsupervised manner, particularly given the abundance of unlabeled data available. They focus on Unsupervised Deep Metric Learning (UDML), a field dedicated to harnessing this unlabeled data to acquire semantic similarity information. UDML typically relies on generating discrete pseudo-labels through clustering techniques, which can be noisy and inconsistent, posing a challenge in learning similarity effectively.

To address this issue, the researchers suggest modeling the data as a piecewise linear manifold, in contrast to treating it as a set of clusters. Each linear manifold approximates the local structure of data in a small neighborhood around a point. Linear manifolds offer greater label homogeneity compared to traditional clusters and enable the estimation of less noisy, continuous-valued similarity by measuring the fit error of the linear manifold to a point. Additionally, the researchers demonstrate that proxies, commonly employed in supervised metric learning, can also be utilized to model the piecewise linear manifold, even in an unsupervised setting. This inclusion contributes to improved performance and faster convergence.

The reviewers thought the paper was interesting and had good improvements. They raised various technical concerns many of which seem to have been addressed. Therefore I recommend acceptance but encourage the authors to address the remaining concerns in their final manuscript.

---

### Decision · Program_Chairs · 2023-11-19

**Decision:**

Accept (Oral)

**Comment:**

After a careful review of the paper titled "Learning Semantically Meaningful Representations in Unsupervised Deep Metric Learning," the reviewers found the research to be interesting and noted significant improvements. The paper addresses the important problem of learning semantically meaningful representations from unlabeled data, specifically focusing on Unsupervised Deep Metric Learning (UDML) and the challenges posed by noisy pseudo-labels generated through clustering techniques.

The proposed approach of modeling data as piecewise linear manifolds is seen as a novel and promising alternative to traditional clustering methods. It offers advantages in terms of label homogeneity and the estimation of less noisy, continuous-valued similarity. The inclusion of proxies, a technique commonly used in supervised metric learning, is also deemed valuable in an unsupervised setting, leading to improved performance and faster convergence.

Given the positive feedback from the reviewers, We decided to accept the paper. However, there are some remaining technical concerns that the authors should address in their final manuscript. It is crucial for the authors to carefully consider and address these concerns to ensure the quality and rigor of the work.

The action PC chair for this paper is Qing Qu, who made the decision after carefully reading the paper as well as the comments by all reviewers and AC. The decision is agreed upon by all PC chairs.